# Biomechanical Comparison of Conventional Plate and the C-Nail^®^ System for the Treatment of Displaced Intra-Articular Calcaneal Fractures: A Finite Element Analysis

**DOI:** 10.3390/jpm13040587

**Published:** 2023-03-27

**Authors:** Roxana Maria Pînzaru, Silviu Dumitru Pavăl, Mihaela Perțea, Ovidiu Alexa, Paul Dan Sîrbu, Alexandru Filip, Adrian Claudiu Carp, Liliana Savin, Norin Forna, Bogdan Veliceasa

**Affiliations:** 1Department of Orthopaedics and Traumatology, Surgical Science (II), Faculty of Medicine, “Grigore T. Popa” University of Medicine and Pharmacy, 16, University Street, 700115 Iasi, Romania; 2Department of Computer Science and Engineering, “Gheorghe Asachi” Technical University, 27, Dimitrie Mangeron, 700050 Iasi, Romania; 3Department of Plastic Surgery and Reconstructive Microsurgery, Surgical Science (I), Faculty of Medicine, “Grigore T. Popa” University of Medicine and Pharmacy, 16, University Street, 700115 Iasi, Romania

**Keywords:** calcaneal fracture, finite element analysis, biomechanics, interlocking nail, C-Nail, locking plate

## Abstract

The C-Nail^®^ system is a novel intramedullary fixation method for displaced intra-articular calcaneal fractures. The aim of this study was to evaluate the biomechanical performance of the C-Nail^®^ system and compare it with conventional plate fixation for the treatment of displaced intra-articular calcaneal fractures using finite element analysis. The geometry of a Sanders type-IIB fracture was constructed using the computer-aided design software Ansys SpaceClaim. The C-Nail^®^ system (Medin, Nové Mesto n. Morave, Czech Republic) and the calcaneal locking plate (Auxein Inc., 35 Doral, Florida) and screws were designed according to the manufacturer specifications. Vertical loading of 350 N and 700 N were applied to the subtalar joint surfaces to simulate partial weight bearing and full weight bearing. Construct stiffness, total deformation, and von Mises stress were assessed. The maximum stress on the C-Nail^®^ system was lower compared with the plate (110 MPa vs. 360 MPa). At the bone level the stress was found to have higher values in the case of the plate compared to the C-Nail^®^ system. The study suggests that the C-Nail^®^ system can provide sufficient stability, making it a viable option for the treatment of displaced intra-articular calcaneal fractures.

## 1. Introduction

Calcaneus fractures account for 1–4% of all fractures (60% of all tarsal fractures) [1], with 60–75% of them being displaced intra-articular calcaneal fractures (DIACFs) [2,3]. Non-surgical treatment of patients with DIACFs frequently has poor functional outcomes due to secondary subtalar arthritis or malunion leading to variable periods of disability [4,5]. The management of these fractures is complex and there are different opinions about the optimal surgical method. Open reduction and internal fixation (ORIF) with plate and screws using an extended lateral approach is currently considered the gold standard treatment in DIACFs [6,7]. This approach is frequently associated with complications, such as major wound-healing problems (5.8–43%), including haematoma, wound-edge necrosis, superficial and deep wound infections, peroneal tendon problems and cutaneous nerve injury [8,9,10].

In order to reduce these complications and facilitate early recovery, surgeons have shifted their focus towards minimally invasive techniques, including percutaneous pinning or screw fixation, external fixator combined with limited internal fixation (EFLIF), the sinus tarsi approach, the limited posterior approach, and arthroscopic-assisted fixation [11,12,13,14,15,16,17,18]. Intramedullary nail fixation has been recently included in the minimally invasive treatment of DIACFs. The C-Nail^®^ and Calcanail^®^ are the most common intramedullary fixation systems used [19,20,21].

The aim of this study was to compare the biomechanical performance and stability of a conventional calcaneal plate in comparison to the C-Nail^®^ system for the treatment of DIACFs using finite element analysis (FEA).

## 2. Materials and Methods

### 2.1. Geometry Reconstruction

The right foot of a female patient was scanned by computed tomography (Siemens Somatom Emotion) for the reconstruction of calcaneal bony structure. The subject was 30 years old, 175 cm tall and weighed 70 kg. The patient agreed to have the scan used in this research. The scan was taken in the transverse direction at 0.6 mm intervals and a pitch factor of 0.8.

The geometry of a Sanders type-IIB fracture was then constructed by using the computer-aided design software Ansys SpaceClaim. The original CT scan was imported into SpaceClaim, cleansed and the fracture inserted. A gap of 0.5 mm was added between the fractured elements. Three major cuts were made according to Blake et al. [22] (see Figure 1):from the midpoint of the anterior facet to the medial calcaneal tuberosity;a vertical cut near the angle of Gissane extending from the lateral cortex to the first cut;a vertical cut separating the posterior tuberosity from the posterior facet.

The C-Nail^®^ implant (Medin, Nové Měston. Moravě, Czech Republic) has a 8 mm diameter and 65 mm long hollow nail which can be extended and closed by an end cap with sizes: 0 mm; 5 mm or 10 mm. Figure 2 shows the C-Nail^®^ implant designed in SpaceClaim according to the manufacturer specifications. The screws were simplified and modelled as cylinders with 3.5 mm diameter. This implant was inserted into the fractured calcaneal bone, as previously detailed.

The second calcanail implant was realized via a calcaneal locking plate (Auxein Inc., Doral, FL, USA) and screws with simplified cylinders with 3.5 mm diameter (Figure 3).

### 2.2. Material Properties and Mesh Creation

The material properties of the finite element models (bone and implants) were assigned according to the values listed in Table 1.

Tetrahedral finite elements were created by Ansys on the calcaneus bone and implants. The first system of calcaneus bone with the C-Nail^®^ implant was meshed with 336,293 elements and 580,519 nodes. The second system of calcaneus bone with the locking plate was meshed with 239,988 elements and 420,096 nodes.

### 2.3. Boundary and Loading Conditions

The friction coefficient between the bone and implant was assigned a value of 0.3, except for the bone–screw interaction which was fully bonded. Two loads of approximately 350 N and 700 N were applied vertically to the surface of the subtalar joint (Figure 4) in order to simulate full stance and single stance standing positions, respectively. In Ansys, we defined the stress force by using absolute loading values on the X,Y and *Z* axes (absolute component values) with the force distributed by using a surface effect (as opposed to direct force applied at one point only). For the 700 N force, the corresponding X,Y and *Z* values are: −100 N, −100 N and −686 N, respectively. In the case of 350 N force, the absolute component values are set to half of the previously stated values.

The ends of the calcaneus bone were fully constrained, as seen in Figure 5. In finite element analysis (FEA) a fully constrained surface is a surface on which all degrees of freedom (DOF) are restricted. This means that the nodes (or points) on the surface cannot move in any direction (such as translation or rotation) in response to the applied loads or boundary conditions.

All other areas of the bone not fully constrained were set to be of the elastic support-type in Ansys. For the volume delimited by the elastic area, we set a foundation stiffness of $10^{9}$ N/m${}^{3}$.

## 3. Results

In this section we evaluate the stress levels and total displacement of bone and implant materials under both single-stance and full-stance loads. To achieve this, we used the von Mises stress calculation obtained from the finite element analysis process performed by Ansys. The von Mises stress is a measure of the equivalent stress in a material, taking into account the combined effects of the normal and shear stresses on a material. This is also known as the “equivalent stress” or “average stress”.

The simulation results are presented in the upcoming subsections using pictures of a colour mesh overlay on the bone surface to aid visualization. The colour codes are visualized in Figure 6 and Figure 7 where the red/warm colours represent higher values of stress and deformation while blue/cold colours represent lower values of stress and deformation. For the specific absolute maximum values measured during the simulation we used numeric representation found in the tables inserted in their respective subsections.

### 3.1. Stresses on Calcaneal C-Nail^®^ Implant

The FEA simulations show greater stresses occur on the C-Nail^®^ main rod at the first fracture line and junction between the nail and screw located in the anterior process, as seen in Figure 8 (red areas). The magnitude of the stresses at these locations in the case of the single-stance standing position was about 110 MPa. In the same case, the average stress on the C-Nail^®^ structure was 1.8 MPa.

### 3.2. Stresses on Calcaneal Plate Implant

The stress in the case of the plate implant was found to be higher with a maximum value of 360 MPa in the red areas seen in Figure 9. On the other hand, the average stress was lower, with a value of 0.1 MPa. These values were for the single-stance standing position. The simulation shows the distribution of stress to be uneven for the plate implant, with parts of the structure heavily loaded while others are minimally loaded.

### 3.3. Stresses on Calcaneal Bone

At the bone level the stress was found to be higher in the case of the plate implant compared to the C-Nail^®^ implant. Figure 10 and Figure 11 show the von Mises stress on the calcaneal bone with the C-Nail^®^ implant and plate implant, respectively.

The C-Nail^®^ implant maximum stress on bone was 28 MPa while the plate implant was 65 MPa. This is again for the case of single stance standing (maximum force).

Figure 12 and Figure 13 show a different view point for the same C-Nail^®^ implant and plate implant cases, respectively.

### 3.4. Displacement on Calcaneal Bone

Calcaneal bone displacement measures the movement or shifting of the calcaneus when force is applied. Lower displacement values are better as displacement usually results in pain, difficulty walking and difficulty bearing weight on the affected foot.

During our simulations we found the C-Nail^®^ system (see Figure 14) to have displacements approximately 10 times lower than the conventional plate system (see Figure 15), under same load conditions.

Table 2 details the exact maximum stress and displacement values for single-stance and full-stance standing positions observed during the simulation.

## 4. Discussion

The use of the C-Nail^®^ system for treating DIACFs has been recently introduced as a treatment method [23]. Restoration of the posterior calcaneal facet is obtained through a sinus tarsi approach and fixed with two lag screws in order to achieve anatomical reduction. After reducing all fragments to the articular block, the calcaneal fracture fixation is performed percutaneously with an interlocking nail, the C-Nail^®^, which results in the restoration of the overall shape of the calcaneus. This facilitates the rapid recovery of the patient and reduces the risks of complications associated with the extended lateral approach. In recent literature, clinical studies have shown favourable results in cases of minimally invasive approaches combined with intramedullary nail (C-Nail^®^) fixation [24]. With the use of minimally invasive techniques the rate of wound-healing complications decreased from 24.9% in the extended lateral approach to 4.9% in the sinus tarsi approach [24,25,26]. The studies of Zwipp et al. [19] and Pompach et al. [27] obtained a low incidence of wound-edge necrosis (1.9%) and deep infection (0.9%). Another study conducted by Veliceasa et al. [24] reported a low rate of complications, with wound-edge necrosis appearing in 4% of the patients, 1.3% developed a superficial infection, and no deep infections.

From a biomechanical perspective, the structural design of the C-Nail^®^ system contributes to better axial stability, while the calcaneus conventional plate provides better lateral support. This innovative system provides angular stability and firm fixation of the bone fragments [28]. The results of this study show that the C-Nail^®^ system improves the stability of the fixed fracture and has the potential to facilitate consolidation.

In 2016 Reinhardt et al. [29] conducted a biomechanical study on 21 cadavers, comparing a polyaxial interlocking plate, the C-Nail^®^ system and the Calcanail^®^. The authors created a Sanders type-IIB fracture model and started biomechanical tests with a vertical load of 1000 N, increasing to 2500 N, and then tested the load to failure at a maximum load of 5000 N. Only one specimen in the C-Nail^®^ group reached the maximum load to failure at 5000 N. The highest load to failure, with a mean of 2808 N (±973.6 N), was in the C-Nail^®^ group, followed by the Rimbus plate group, with a mean of 2041 N (±603.6 N), and the Calcanail^®^ group, with a mean of 1751 N (±756.3 N). In our finite element analysis, the vertical load applied on the subtalar joint was 350 N and the results support the C-Nail^®^ as the calcaneal implant with a higher stress resistance and better efficiency in the treatment of DIACFs in comparison with the plate. We obtained a peak bone stress of 56 MPa in the C-Nail^®^ system, while in the plate implant was 185 MPa.

A finite element analysis of Ni et al. [30] compared the plate fixation with the Calcanail^®^ and a modified version of it with a 3.5 mm diameter transfixation screw for a Sanders type-IIIAB calcaneal fracture. They applied a vertical load of 700 N to the subtalar joint and compared the construct stiffness, fracture migration and maximum stress of the implant and calcaneus. In the plate construct, the maximum stress was 102.68 MPa and in the modified Calcanail^®^ the lowest was 84.78 MPa. The peak von Mises stress was also lower in the modified Calcanail^®^ construct [30]. Under similar loading conditions, our study supports the fact that a minimally invasive centromedullar implant has lower stress values and thus a higher overall resistance. The maximum stress on the C-Nail^®^ system was 110 MPa, while on the conventional plate was 360 MPa.

Stress distribution is an important indicator in hardware failure and the stress magnitude is relevant to the safety of the implant under given loading conditions when validated by practical biomechanical tests. In this study, the maximal load applied vertically on the subtalar joint was 700 N. The C-Nail^®^ system experienced the least stress, distributed between the nail and screws, with minimal effects on the model. The maximum stress in the C-Nail^®^ system occurs on the nail, at the level of the first fracture line and the junction between the nail and screw located in the anterior process. The fixation with the locked plate revealed the highest stress, concentrated in the anterior and central branches of the plate extending across the fracture line, as well as in the plate–screw junction. Yu et al. [31] measured the stress on the screw in a finite element model of a Sanders type-IIB fracture fixed with a conventional plate and an anatomical plate and their results are similar to our analysis. The stress observed on the screws is higher on the screws close to the fracture line than those apart from it. It is important to note that the applied forces did not exceed the maximum resistance of the material from which the plate is made, stainless steel (up to 600 MPa). This may suggest that the implant could be considered safe, with a small chance of failure with loading forces equivalent to full weight bearing. These results are similar with the ones reported by Ouyang et al. [32].

To the best of our knowledge, this is the first study comparing the biomechanical performance of a conventional calcaneal plate with the C-Nail^®^ system for DIACFs using finite element analysis (FEA).

One of the strengths of our study is the mesh creation. The calcaneus bone and locking plate were meshed with 239,988 elements and 420,096 nodes, while the calcaneus bone with the C-Nail^®^ implant were meshed with 336,293 elements and 580,519 nodes. The number of elements and nodes is important in FEA because it impacts the accuracy of the simulation. Usually a greater number of elements and nodes provides a more detailed representation of the deformation and displacement of the model. A higher number of nodes allows a more accurate representation of the displacement field and can capture more localized behaviours. However, there are limitations imposed by the computational power as a higher number of elements will require a greater number of computations to be made.

There are still some limitations of this study that should be noted. Firstly, only a Sanders type-IIB fracture was simulated and tested with the FEA. Further research, both biomechanical and clinical, concerning other types of calcaneal fractures should be carried out in future. Secondly, the soft tissue envelope was excluded from the model and axial loads were applied only at the level of the posterior facet as in the other tests that were previously made. Nevertheless, we believe that the used model is reliable in testing the two fixation’s biomechanical conduct.

## 5. Conclusions

The C-Nail^®^ system provided greater stability than the conventional calcaneal plate in the treatment of DIACFs. The C-Nail^®^ system fixation combined with the low complication rate of the sinus tarsi approach represents a viable option for the treatment of DIACFs. The results of this study may provide surgeons with useful information when choosing the optimum implant for the fixation of DIACFs.

## Figures and Tables

**Figure 1 jpm-13-00587-f001:**
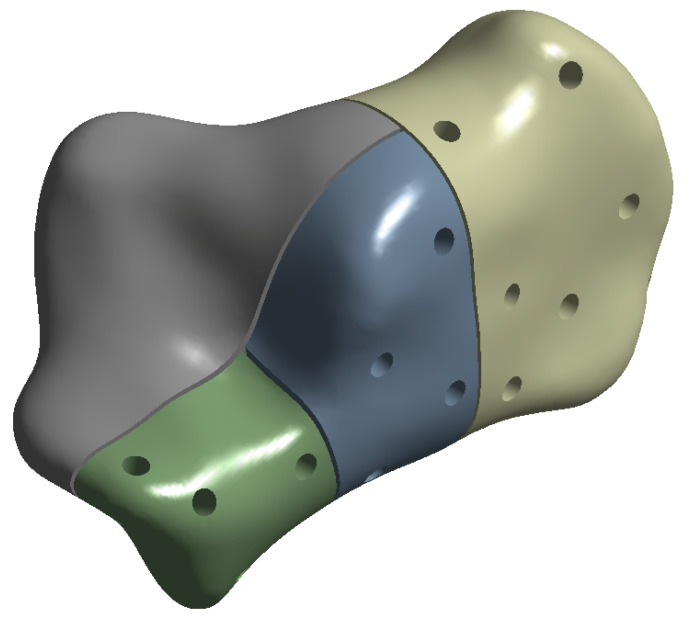
Sanders fracture type-IIB.

**Figure 2 jpm-13-00587-f002:**
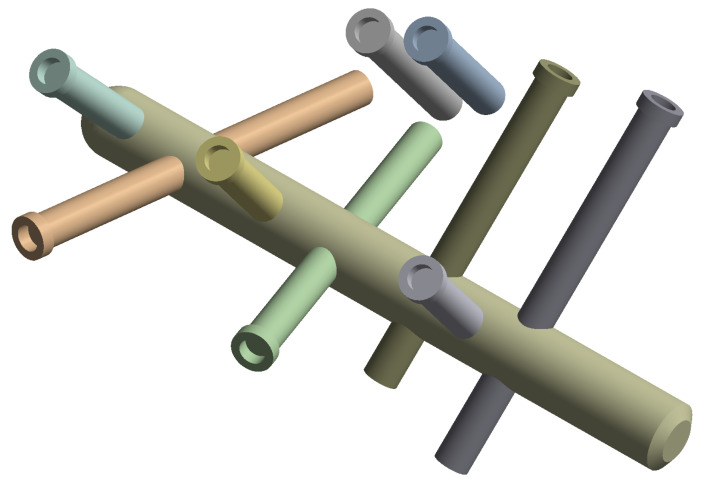
C-Nail^®^ implant.

**Figure 3 jpm-13-00587-f003:**
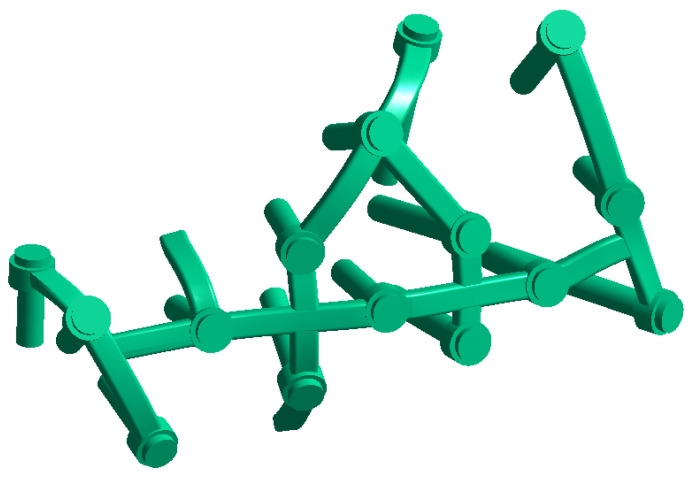
Calcaneal locking plate.

**Figure 4 jpm-13-00587-f004:**
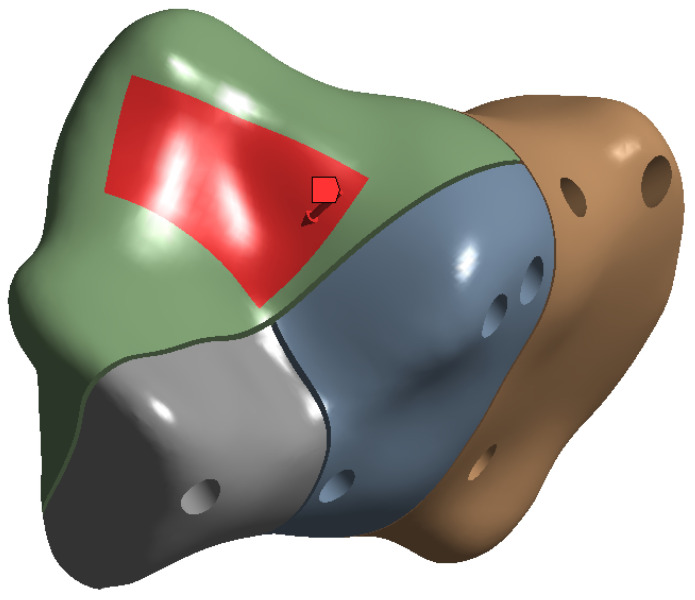
Force distribution and orientation.

**Figure 5 jpm-13-00587-f005:**
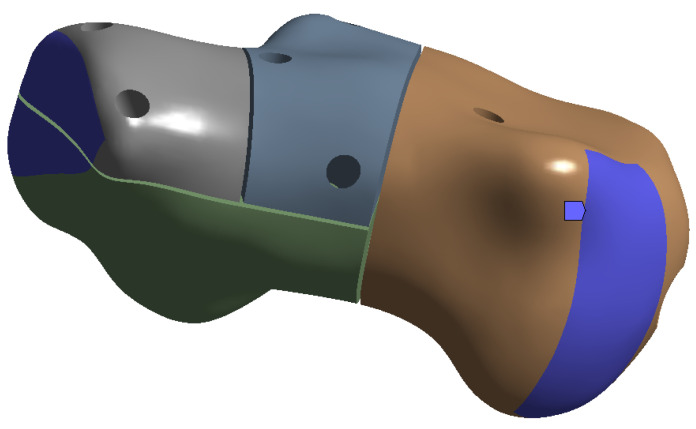
Fixed boundaries (coloured in dark blue).

**Figure 6 jpm-13-00587-f006:**
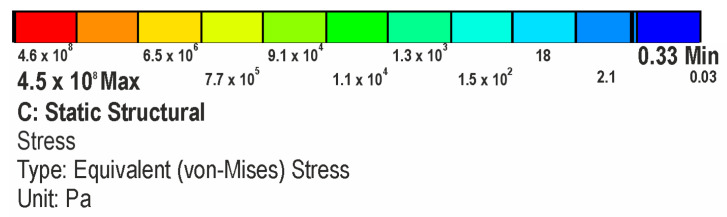
Colour-coded scale for deformation simulations.

**Figure 7 jpm-13-00587-f007:**
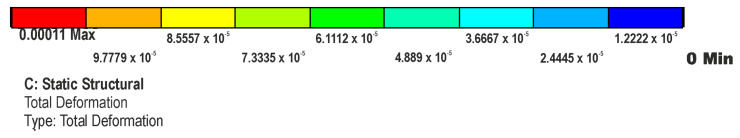
Colour-coded scale for stress simulations.

**Figure 8 jpm-13-00587-f008:**
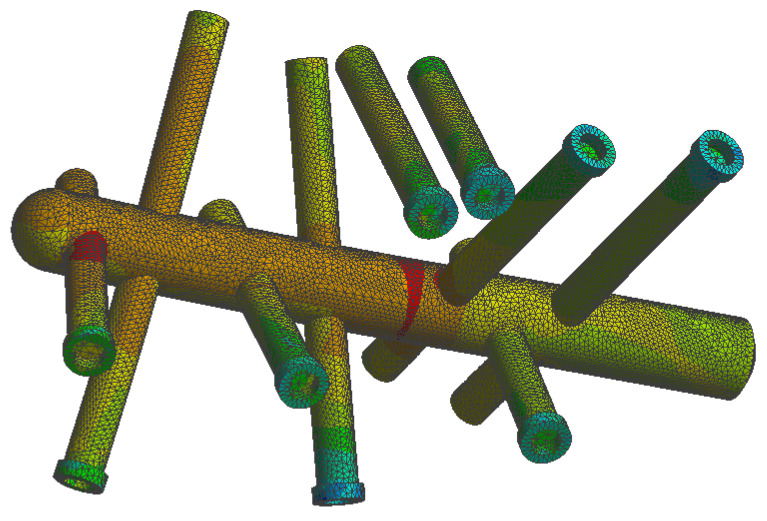
The von Mises stress on the C-Nail^®^ implant structure.

**Figure 9 jpm-13-00587-f009:**
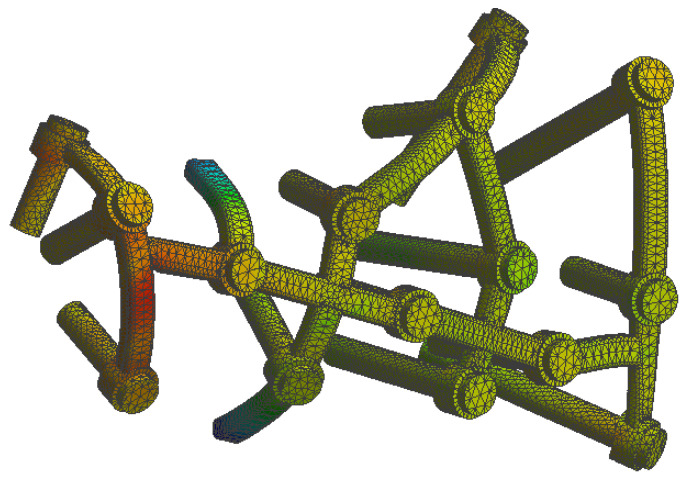
The von Mises stress on the plate implant structure.

**Figure 10 jpm-13-00587-f010:**
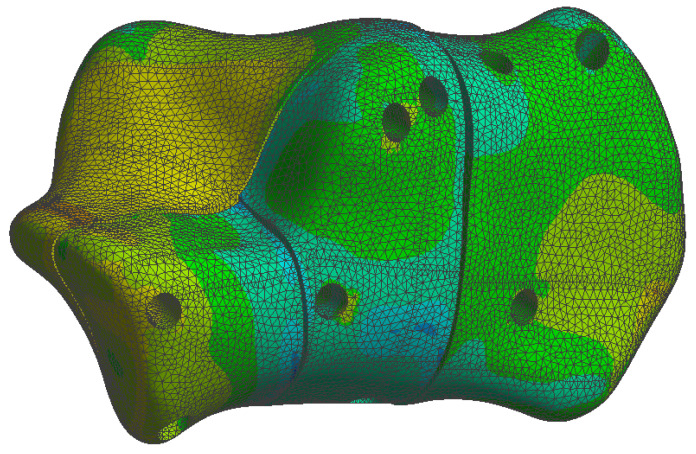
The von Mises stress for calcaneal bone with the C-Nail^®^ implant (left lateral view).

**Figure 11 jpm-13-00587-f011:**
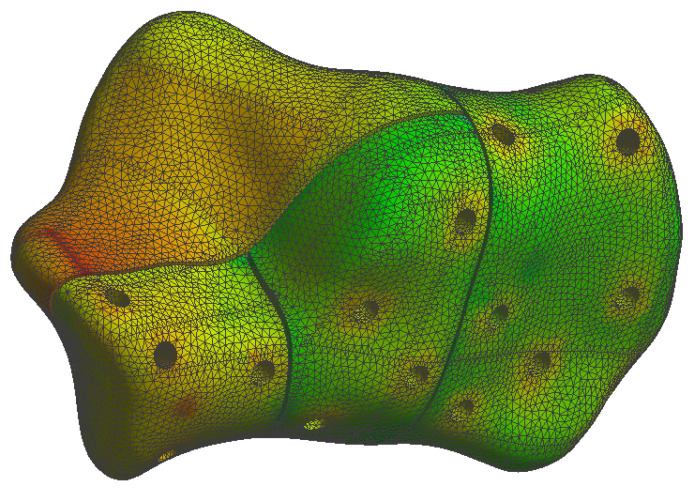
The von Mises stress for calcaneal bone with the plate implant (left lateral view).

**Figure 12 jpm-13-00587-f012:**
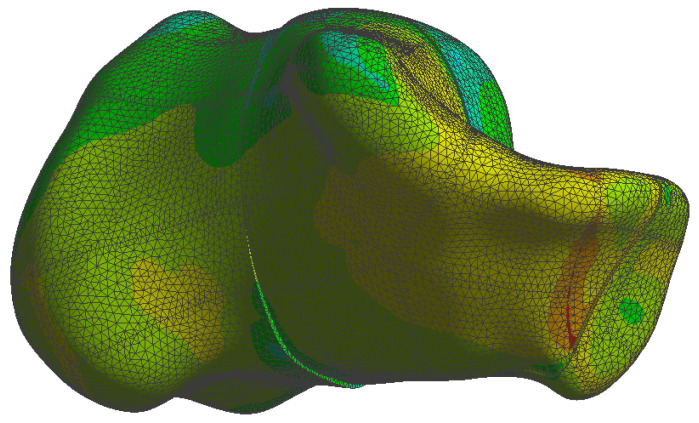
The von Mises stress for calcaneal bone with the C-Nail^®^ implant (left medial view).

**Figure 13 jpm-13-00587-f013:**
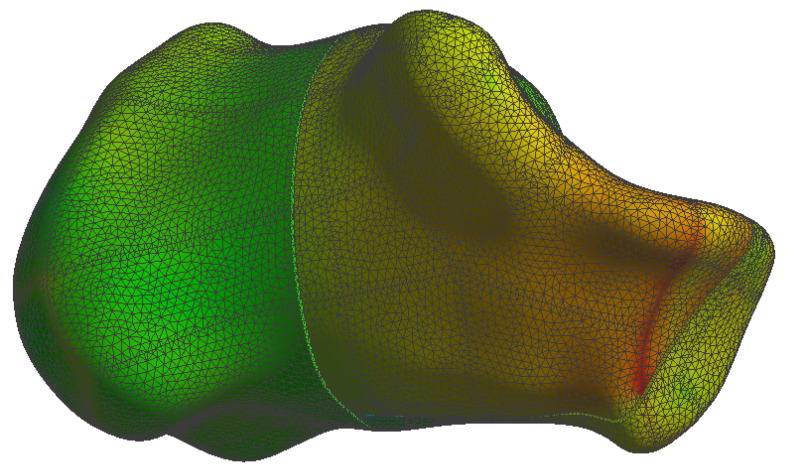
The von Mises stress for calcaneal bone with the plate implant (left medial view).

**Figure 14 jpm-13-00587-f014:**
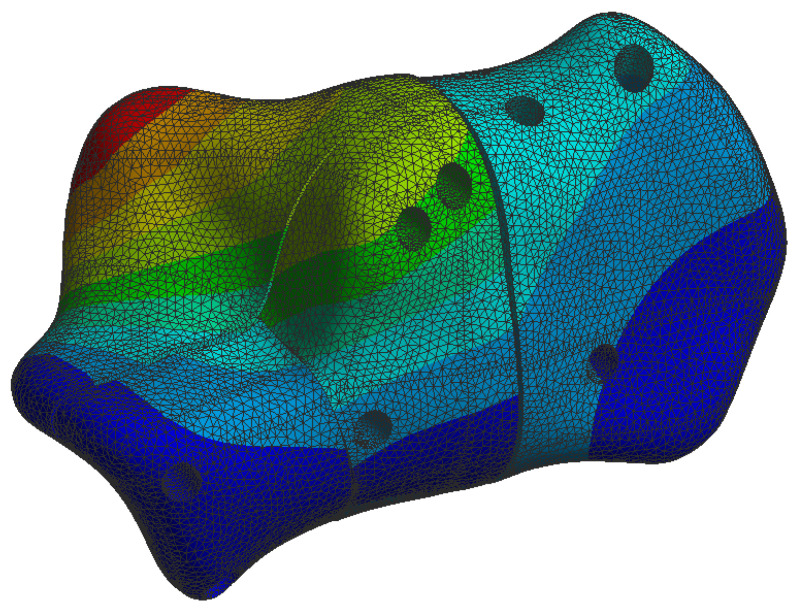
Calcaneal bone displacement in the case of the C-Nail^®^ simulation.

**Figure 15 jpm-13-00587-f015:**
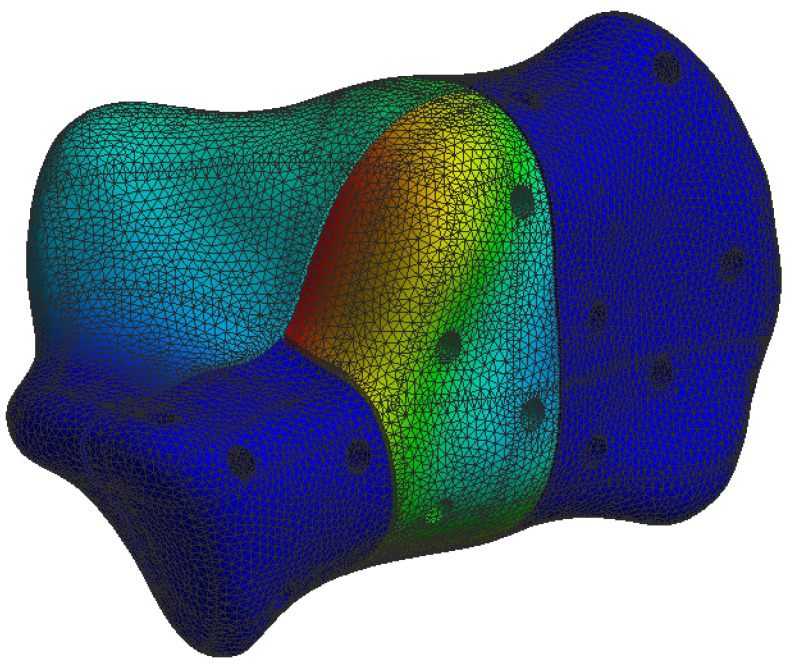
Calcaneal bone displacement in the case of the plate implant simulation.

**Table 1 jpm-13-00587-t001:** Physical characteristics for the finite element models.

Material	Young’s Modulus (MPa) ^1^	Poisson’s Ratio ^2^
Calcaneal bone	1.89 $\times \;10^{4}$	0.3
Steel (implants)	2 $\times \;10^{5}$	0.3

^1^ Young’s modulus is a measure of a material’s stiffness and is used to model the elastic behaviour of materials
such as metals and bone. ^2^ Poisson’s ratio is a measure of a material’s lateral contraction and is used to model the
deformation of materials in response to applied loads.

**Table 2 jpm-13-00587-t002:** Maximum stress and displacement values obtained during the simulations.

	Max von Mises Stress (MPa)	Max Displacement (mm)
**C-Nail^®^ System**		
Single-standing stance (700 N)	110	0.01
Full-standing stance (350 N)	56	0.007
**Calcaneal Plate**		
Single-standing stance (700 N)	360	1.1
Full-standing stance (350 N)	185	0.6

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
