# Peer review of "Biomechanical Comparison of Conventional Plate and the C-Nail® System for the Treatment of Displaced Intra-Articular Calcaneal Fractures: A Finite Element Analysis"

_jpm, 2023, doi:10.3390/jpm13040587_

Round 1

Reviewer 1 Report

The topic of the study is interesting because the authors propose a comparison between two different surgical implants for the treatment of displaced and intra-articular calcaneal fractures.

The manuscript is overall well written. The structure results properly organized

The abstract is comprehensive, and the aim of the study clearly described.

The introduction is overall good

At lines 17-18 authors, if they consider it appropriate, could better specify epidemiology of calcaneus fractures with the following article:10.3390/jcm11195660 “Calcaneus fractures account for 1–4% of all fractures (60% of all tarsal fractures)”.  Furthermore, among the different surgical techniques used and found in literature, authors could mention this article. doi10.3390/jcm12010020

The methodological approach is correct. Sub-sections are widely and carefully described. Images, charts, and descriptions are clear and comprehensive. 

Results are well presented

Discussion is globally acceptable; references are well reported. 

Conclusions are well written. It would be interesting to conduct further studies to evaluate the use of C-nail also in the other Sanders-type fractures.

Author Response

Dear Revisor,

Thank you for your comments and the opportunity to revise our manuscript entitled Biomechanical comparison of conventional plate and the C-Nail® system for the treatment of displaced intra-articular calcaneal fractures: A finite element analysis submitted to Journal of Personalized Medicine, Special Issue of Personalized Management in Orthopedics and Traumatology. We appreciate the time and effort that you have dedicated to providing your valuable feedback on our manuscript. We would like to thank you for careful and thorough reading of this manuscript and for the thoughtful comments and constructive suggestions, which help to improve the quality of this manuscript.

Regarding your concern: At lines 17-18 authors, if they consider it appropriate, could better specify epidemiology of calcaneus fractures with the following article:10.3390/jcm11195660 “Calcaneus fractures account for 1–4% of all fractures (60% of all tarsal fractures)”.  - Furthermore, among the different surgical techniques used and found in literature, authors could mention this article. doi: 10.3390/jcm12010020. 

Response 1.

  1. We changed at lines 17-18 with the following sentence “Calcaneus fractures account for 1–4% of all fractures (60% of all tarsal fractures)[1]” and added article Cianni, L.; Vitiello, R.; Greco, T.; Sirgiovanni, M.; Ragonesi, G.; Maccauro, G.; Perisano, C. Predictive Factors of Poor Outcome in Sanders Type III and IV Calcaneal Fractures Treated with an Open Reduction and Internal Fixation with Plate: A Medium-Term Follow-Up.  Clin. Med.202211, 5660. https://doi.org/10.3390/jcm11195660 as refference.
  2. Also, at line 30 we added a refference to Caravelli, S.; Gardini, G.; Pungetti, C.; Gentile, P.; Perisano, C.; Greco, T.; Rinaldi, V.G.; Marcheggiani Muccioli, G.M.; Tigani, D.; Mosca, M. Intra-Articular Calcaneal Fractures: Comparison between Mini-Invasive Approach and Kirschner Wires vs. Extensive Approach and Dedicated Plate—A Retrospective Evaluation at Long-Term Follow-Up.  Clin. Med.202312, 20. https://doi.org/10.3390/jcm12010020

We hope that the manuscript changes/replies are in conformity to your expectations. We hope the revised manuscript will better suit the Journal of Personalized Medicine but we are happy to consider further revisions, and we thank you for your continued interest in our research.

Best Regards,

The authors.

Reviewer 2 Report

I think this paper is a great article. Nevertheless, calcaneal nails are still of limited use.
Of course, the authors explained that selection could be induced according to the fracture type.
On the other hand, I wonder if the screws provided in this calcaneal nail set are used for fixing the sustenticulum tali.
Regarding finite element analysis, it has been analyzed with appropriate content, and there is no content to be modified.
Thank you for submitting a good thesis.

Author Response

Dear Revisor,

Thank you for your comments and the opportunity to revise our manuscript entitled Biomechanical comparison of conventional plate and the C-Nail® system for the treatment of displaced intra-articular calcaneal fractures: A finite element analysis submitted to Journal of Personalized Medicine, Special Issue of Personalized Management in Orthopedics and Traumatology. We appreciate the time and effort that you have dedicated to providing your valuable feedback on our manuscript. We would like to thank you for careful and thorough reading of this manuscript and for the thoughtful comments and constructive suggestions, which help to improve the quality of this manuscript.

Regarding your concern: On the other hand, I wonder if the screws provided in this calcaneal nail set are used for fixing the sustenticulum tali.

 Response 1: This study was focussed on a matemathical analisys of the biomechanical properties of the C-nail system vs locked plate in fixing a Sanders IIB fracture.

We did not present the technique of C-Nail fixation, nor the indication and contraindication of the system, these were provided in previous articles [1,2]. One of the major contraindication for c-nail fixation is sustentaculum tali with multiple fragmentation of the medial facet.

On the other hand, the sustentacular fragment is the most constant and, for anatomical restauration of the posterior facet and overall shape of the calcaneus, all the other fragment are reduced to this fragment. Also, the sustentaculum tali is the strongest part of the calcaneus and therefore, it is an optimal spot for screw placement. After anatomical reduction of the calcaneus, the C-Nail system is inserted percutaneously, and the main step is the introduction of the sustentacular screws.

Besides the fact that the screws insertion into the sustentaculum offers the coorect direction (rotation) of the nail into the bone, they offers the advantage of obtaining and maintaing maximum stability of the subtalar joint and prevents later displacement.

  1. Amlang, M., Zwipp, H., Pompach, M., & Rammelt, S. (2017). Interlocking nail fixation for the treatment of displaced intra-articular calcaneal fractures. JBJS Essential Surgical Techniques7(4).
  2. Veliceasa, B., Filip, A., Pinzaru, R., Pertea, M., Ciuntu, B., & Alexa, O. (2020). Treatment of Displaced Intra-articular Calcaneal Fractures With an Interlocking Nail (C-Nail). Journal of orthopaedic trauma34(11), e414-e419.

We hope that the manuscript changes/replies are in conformity to your expectations. We hope the revised manuscript will better suit the Journal of Personalized Medicine but we are happy to consider further revisions, and we thank you for your continued interest in our research.

Best Regards,

The authors.

Reviewer 3 Report

1. The FEA analysis conducted on only one subject, hardly to support the C-Nail® system can provide sufficient stability then conventional plate, I would recommend that adding at least two cases.

2. Materials and Methods should add more detailed about the setting parameter and stress application.

3. In all figure, what is the color 'blue to red' representing?  please adding the scale.

Author Response

Point 1. The FEA analysis conducted on only one subject, hardly to support the C-Nail® system can provide sufficient stability then conventional plate, I would recommend that adding at least two cases.

Response 1.

Dear Reviewer,

Thank you so much for reviewing our paper and for your remarks that improved this article. To answer your first concern, our article is more of a mathematical study where we simulate a fracture and implant on a healthy bone. Preserving these initial conditions the results will not change even if we chose the CT of a second person as the base for simulation. We think in a clinical study having multiple study cases would indeed be a must.

Point 2. Materials and Methods should add more detailed about the setting parameter and stress application.

Response 2. We did add more details referring to the parameters of the FEA method that we used and the application of stress, under the “Boundary and loading conditions” subsection.

Point 3. In all figure, what is the color 'blue to red' representing?  please adding the scale.

Response 3. We added more details about the color codes used in our visualizations that indeed will bring more clarity about the distribution of the stress in our simulation results (please see the modified “Results” section).

We hope that the manuscript changes/replies are in conformity to your expectations. We hope the revised manuscript will better suit the Journal of Personalized Medicine but we are happy to consider further revisions, and we thank you for your continued interest in our research.

Best Regards,

The authors.